# Active rheumatoid arthritis in a mouse model is not an independent risk factor for periprosthetic joint infection

Rishi Trikha, Danielle Greig, Troy Sekimura, Nicolas Cevallos, Benjamin Kelley, Zeinab Mamouei, Christopher Hart, Micah Ralston, Amr Turkmani, Adam Sassoon, Alexandra Stavrakis, Nicholas M. Bernthal*

University of California, Los Angeles, Los Angeles, CA, United States of America

* nbernthal@mednet.ucla.edu

**Data Availability Statement:** All relevant data are within the manuscript and its Supporting Information files.

## Abstract

### Introduction

Periprosthetic joint infection (PJI) represents a devastating complication of total joint arthroplasty associated with significant morbidity and mortality. Literature suggests a possible higher incidence of periprosthetic joint infection (PJI) in patients with rheumatoid arthritis (RA). There is, however, no consensus on this purported risk nor a well-defined mechanism. This study investigates how collagen-induced arthritis (CIA), a validated animal model of RA, impacts infectious burden in a well-established model of PJI.

### Methods

Control mice were compared against CIA mice. Whole blood samples were collected to quantify systemic IgG levels via ELISA. *Ex vivo* respiratory burst function was measured via dihydrorhodamine assay. *Ex vivo Staphylococcus aureus* Xen36 burden was measured directly via colony forming unit (CFU) counts and crystal violet assay to assess biofilm formation. *In vivo*, surgical placement of a titanium implant through the knee joint and inoculation with *S. aureus* Xen36 was performed. Bacterial burden was then quantified by longitudinal bioluminescent imaging.

### Results

Mice with CIA demonstrated significantly higher levels of systemic IgG compared with control mice (p = 0.003). *Ex vivo*, there was no significant difference in respiratory burst function (p = 0.89) or *S. aureus* bacterial burden as measured by CFU counts (p = 0.91) and crystal violet assay (p = 0.96). *In vivo*, no significant difference in bacterial bioluminescence between groups was found at all postoperative time points. CFU counts of both the implant and the peri-implant tissue were not significantly different between groups (p = 0.82 and 0.80, respectively).

### Conclusion

This study demonstrated no significant difference in *S. aureus* infectious burden between mice with CIA and control mice. These results suggest that untreated, active RA may not

**Funding:** This work was supported by the National Institute of Arthritis and Musculoskeletal and Skin Diseases of the National Institutes of Health, Award Number 5K08AR069112-01. The funders had no role in study design, data collection and analysis, decision to publish, or preparation of the manuscript.

**Competing interests:** The authors have declared that no competing interests exist.

represent a significant intrinsic risk factor for PJI, however further mechanistic translational and clinical studies are warranted.

## Introduction

Periprosthetic joint infection (PJI) is a challenging complication faced by orthopaedic surgeons and is the most common cause of implant failure following arthroplasty [1, 2]. PJI is associated with significant morbidity and mortality, with a higher 5-year mortality than both breast and prostate cancer [3]. Furthermore, the treatment of PJI places a tremendous economic burden on the United States healthcare system with annual hospital costs projected to be $1.85 billion by 2030 [4–8]. Once colonized, an orthopaedic implant is deemed unsalvageable as a biofilm associated infection is recalcitrant to both systemic therapies and surgical interventions. Thus, it is important to understand factors that may alter infectious risk so that we may prevent PJI as the number of total joint arthroplasties and subsequent PJIs are projected to increase multiple fold in the United States in the next decade [4, 9–12].

As highlighted by the American Association of Hip and Knee Surgeons (AAHKS) and the American College of Rheumatology (ACR) [13], there is an unmet demand for thorough investigation of the perioperative infectious risk associated with rheumatoid arthritis (RA) following arthroplasty. RA is the most common autoimmune inflammatory arthropathy, affecting 0.5% of the general population [14–16]. RA is characterized by a dysregulated innate immune response, an upregulated adaptive immune response against self-antigens, disordered cytokine (TNF-α, IL-1 and IL-6) activation, increased oxidative stress, and dysregulated osteoclast and chondrocyte activation [17–22]. This often results in the auto-reactive T cell-mediated destruction of articular cartilage via synovial hypoxia as well as reactive oxidative species generation that can lead to debilitating joint pain ultimately necessitating arthroplasty [18, 23–25]. Although there are numerous studies on novel therapeutic targets to counter the synovial inflammation associated with RA [25–27], the infectious burden of RA is not yet clear. Some retrospective studies have demonstrated that patients with RA that undergo total joint arthroplasty have a higher incidence of PJI than patients with osteoarthritis (OA) alone [24, 28–33]. Other studies, however, suggest that no such difference in infection rate exists between patients with RA and those with OA following total hip or knee arthroplasty [34–36]. Furthermore, as many of the immunosuppressive medications used to treat RA carry risk for infection, it is unknown whether the immune dysregulation of RA alone, independent of medications, carries infectious risk. This distinction is of profound clinical significance as the medication-related risk may be modifiable in the perioperative period whereas the intrinsic risk of the underlying disease is far more challenging to mitigate. The paucity of evidence as to the mechanistic etiology behind any purported increased infectious risk in patients with RA [13, 37], along with the relative uncertainty of whether such an infectious risk truly exists, emphasizes the need for further research.

In an effort to more clearly elucidate this infectious risk, the current study took a novel approach by combining two well-validated models of RA as well as PJI. Collagen-induced arthritis (CIA) is the gold standard animal model of RA and forms the genetic and molecular background of the majority of genetically modified RA strains in C57BL/6 mice. CIA is characterized by T-cell and cytokine-dependent sustained responses to exogenous type II collagen that causes recruitment of macrophages, neutrophils and lymphocytes to the synovium [22, 38]. The susceptibility of mice to CIA is dependent on the MHC H-2 mouse haplotype which is similar to RA in humans where the MHC molecule, HLA-DR drives RA severity [39]. CIA

has thus been repeatedly shown to be a clinically, radiographically, immunologically, and pathogenically well-validated and reproducible model of RA [38, 40–45].

Given the multifactorial nature and challenges associated with a clinical study of patients with RA undergoing total joint arthroplasty, this study aimed to interrogate a well-validated mouse model [46–49] of PJI as well as a well-validated model of RA [38, 40–45] to determine how RA alone affects perioperative infectious burden. Elucidating, isolating and quantifying any purported perioperative infectious risk of RA as well as further investigating its modifiability is essential to formulating strategies to decrease the incidence of PJI in this patient population.

## Materials and methods

### Ethics statement

All practices were set forth by federal regulations detailed in the Animal Welfare Act (AWA), PHS Policy for the Humane Care and Use of Laboratory Animals, the 1996 Guide for the Care and Use of Laboratory Animals as well as UCLA (University of California, Los Angeles) policies by procedures in the UCLA Animal Care and Use Training Manual. The UCLA Chancellor's Animal Research Committee (ARC) approved all animal research (ARC# 2008-112-41). Continuous inhalation isoflurane (2%) was the anesthetic used during all experiments. No animals became ill during the study and all causes of death were due to planned sacrifice at the end of each experiment via inhalation isoflurane (2%) followed by cervical dislocation. Adequate sample size for each arm of the experiment was based on statistical analyses from prior studies designed to determine power [47, 49].

### Bacterial strain

*Staphylococcus aureus* Xen36 (PerkinElmer, Hopkinton, MA) is a derivative of a parent strain, ATCC-29525 (Wright), that contains a bioluminescent luxABCDE operon genomically integrated into a stable bacterial plasmid. When bacteria are metabolically active, a blue-green light with a maximal emission wavelength of 490nm is produced. This bioluminescent strain has been previously validated as the optimal strain with which to longitudinally monitor *S. aureus* burden in an implant-associated infection model [46–49].

Bacteria was prepared according to recently published protocols [46–51]. Briefly, *S. aureus* Xen36 was first streaked onto tryptic soy broth (TSB) agar kanamycin plates (TSB plus 1.5% Bacto agar; BD Biosciences) and grown for approximately 24 hours at 37˚C. Due to the presence of a kanamycin-resistant marker integral to the lux operon, this step removes potential contaminants. Single colonies were subsequently isolated and cultured in TSB with 200ug/mL kanamycin in a shaking incubator (196rpm) at 37˚C for 16 hours (MaxQ 4,450, Thermo Fisher Scientific, Canoga Park, CA). An additional two-hour subculture was performed in order to obtain bacteria in mid-logarithmic phase growth. Following centrifugation, cells were pelleted, resuspended and washed in Phosphate-Buffered Saline (PBS). Bacteria was diluted with PBS to its final concentration of 1 x $10^3$ *S. aureus* Xen36 CFUs/2 μL, which was quantitated via spectrophotometry (OD, 600nm; Biomate 3; Thermo Fisher Scientific) and subsequently inoculated during the surgical procedure.

### Induction of collagen-induced arthritis

Eight week-old C57BL/6 wildtype male mice (Jackson Laboratories, Bar Harbor, ME) were housed and stored with a 12-hour light and dark cycle with free access to water and a standard pellet diet. Veterinary staff monitored mice daily to verify their health throughout the duration

of the study. A maximum of four mice were permitted to be housed in each cage. All mice were housed in the same room to minimize confounding.

The protocol for induction of CIA published by Inglis et al. was used on eight week old C57BL/6 mice as these species of mice are indeed susceptible to CIA induced by chicken type II collagen [41]. Briefly, a mixture of chicken type II collagen dissolved in 0.1M acetic acid was mixed with an equal volume of complete Freund's adjuvant (all reagents purchased from Sigma-Aldrich, Saint Louis, MO, USA). Eight week-old mice were then anesthetized using isoflurane (2%) and 0.1 mL of the resulting solution was injected at the base of the tail of each mouse. A booster injection was given two weeks following the primary immunization. Mice were randomized either to receive the immunization (CIA group) or sterile saline (control group). Mice were clinically evaluated six weeks after the primary immunization, as time to arthritis onset is between three and six weeks [41]. The paws of each mice in the CIA group were given a score of 0, 1, 2, or 3 based on the amount of dorsal edema, with 0 correlating with no edema and 3 correlating with severe edema. A sum total score was generated for each mouse, with a maximum score of 12. As the incidence of arthritis after immunization is reportedly 50–75% [41], only mice that clinically showed any edema on the dorsum of their paws six week after induction were selected for *ex vivo* and *in vivo* experiments. All mice were fourteen weeks of age at the time of experiments.

### Ex vivo mouse IgG ELISA to confirm systemic effect of CIA

In order to confirm an altered immunologic profile in CIA mice, a mouse IgG ELISA (Bethyl Laboratories, Montgomery, TX) was performed to quantify systemic IgG levels as anti-cyclic citrullinated peptide antibodies are of IgG isotype and are a specific diagnostic marker of rheumatoid arthritis. Briefly, whole blood was collected from six mice in the CIA group and six mice in the control group via cardiac puncture under 2% isoflurane inhalation anesthesia. Ethylenediaminetetraacetic acid (EDTA) was added to blood samples in a 1:10 ratio to prevent coagulation. 100 μL of blood from each mouse was added to mouse IgG pre-coated wells and mixed with 100 μL of anti-IgG Detection Antibody. Streptavidin-conjugated horseradish peroxidase was then added followed by 3,3',5,5'-tetramethylbenzidine. The colorimetric reaction was then terminated with 100 μL of 0.3 M sulfuric acid and the absorbance of the yellow resultant product was measured for each sample at 450nm using a plate reader (FLUOstar Omega, BMG Labtech, Ortenberg Germany). Per Bethyl Laboratories protocol, a four-parameter curve was subsequently generated using standardized concentrations of IgG. This curve was interpolated to allow for quantitation of IgG concentration in the CIA mice blood samples as well as control samples.

### Ex vivo quantification of respiratory burst

Blood was collected as above from six mice in each experimental group. 100 μL of blood was added from each mouse to each well within a 96-well flat bottom plate (Corning Costar, Corning, New York). A dihydrorhodamine (DHR) 123 assay was used to assess reactive oxygen species (ROS). Briefly, 10 μL of DHR 123 Assay Reagent, 25 μL of Phorbol myristate acetate and two 2mL of Red Blood Cell Lysis Buffer was added to each plate. An excitation filter of 485nm and an emission filter of 520nm was used to quantitate mean fluorescent intensity using a fluorescent plate reader (FLUOstar Omega, BMG Labtech, Ortenberg, Germany).

### Ex vivo CFU quantification of *S. aureus* after interaction with whole blood

Six mice in each group underwent whole blood collection as above. 10μL of 1 x $10^3$ *S. aureus* Xen36 CFU/mL and 10μL of blood was gently vortexed and incubated for one hour at 37˚C.

20μL of the resulting solution was subsequently spread onto a TSB agar plate (tryptic soy broth [TSB] plus 1.5% Bacto agar; BD Biosciences) and incubated for 16 hours at 37°C. CFUs were counted for each plate and expressed as CFUs/mL.

## Ex vivo quantification of biomass of biofilm

Six mice in each group underwent whole blood collection as above. 100 μL of 1 x $10^7$ *S. aureus* Xen36 CFU/mL and 100 μL of blood from each mouse were mixed into each well of a 96-well flat bottom plate. Additionally, 200 μL of saline was used as a standardized control. The plate was incubated for 24 hours at 37°C (MaxQ 4450; ThermoFisher Scientific) to permit sufficient biofilm formation. Wells were then washed with PBS three times to remove blood and non-adherent bacteria. In order to quantify mass of the residual biofilm, a well-validated crystal violet assay (Abcam, Cambridge, United Kingdom) was performed and absorbance was measured by OD at 595nm (FLUOstar Omega, BMG Labtech, Ortenberg Germany). Values were reported as absorbance units.

## In vivo mouse surgical protocol

Twenty-two mice were randomized into the following groups: 2 sterile controls, 8 infected controls (non-immunized), and 12 CIA-immunized. Mice were anesthetized via isoflurane (2%) inhalation. Implant surgery and inoculation was performed as described in prior protocols [46–49]. Briefly, a medial parapatellar approach was used to displace the patella and expose the distal femur. After subsequent reaming of the femoral intramedullary canal with a 25-gauge followed by a 21-gauge needle, an orthopaedic-grade titanium Kirschner wire (0.8mm in diameter, 6mm in length; DePuy Synthes, Warsaw, IN) was inserted in a retrograde fashion with 1mm protruding into the joint space. The protruding wire was then inoculated with either 2μL of sterile saline (0.9% NaCl, sterile control group) or 2μL of 1 x $10^3$ *S. aureus* Xen36 CFUs/mL of bacteria (infected control and CIA groups). 5–0 polyglycolic acid sutures were used to close the surgical site. Following the procedure, all mice underwent high resolution X-rays using the IVIS Lumina X5 (PerkinElmer, Waltham, MA) to ensure appropriate placement of the implant.

## In vivo longitudinal monitoring of bacterial burden and CFU quantification

As previously described [46–49], the IVIS Lumina X5 (PerkinElmer, Waltham, MA) was used to obtain bioluminescent images representative of infectious burden on postoperative days (POD) 0, 1, 3, 5, 7, 10, 14, 18, 21, 25 and 28. Data were quantified as total flux (photons per second per $cm^2$ per steradian [photons/s/$cm^2$/sr]). Mice were sacrificed on POD 28 and bacterial CFU counts of the bacteria adherent to the implant as well as in the surrounding joint tissue were quantified. To isolate adherent bacteria, implants underwent sonication in 500 μL 0.3% Tween-80 (ThermoFisher Scientific) in TSB. To isolate bacteria in the surrounding joint tissue, tissue was homogenized in 1000 μL of PBS with 4 homogenizing beads (Pro200H Series homogenizer; Pro Scientific) and the implants underwent sonication in 500 μL 0.3% Tween-80 (ThermoFisher Scientific) in TSB. Samples from tissue and implants were then drop plated and incubated for 16 hours at 37°C. CFUs were counted and expressed as CFUs/mL.

## Clinical CIA severity scoring

Mice in the CIA group were given a clinical score of 0–12 to describe severity of CIA, as described above. Scores were given by a single observer at all bioluminescent imaging POD

time points. Scores were calculated by noting the amount of swelling on the dorsum of each paw and giving each paw a score of 0–3 [41].

## Statistical analysis

Consistent with prior research utilizing this implant infection model, each experimental group contained at least 6 mice in order to sufficiently power the study [47]. A linear mixed effects regression model was used to analyze bioluminescent and clinical CIA score data. Data were expressed as mean +/- standard error of the mean (SEM). A Student's t-test (one or two-tailed where indicated) was conducted, and significance was defined as a p-value <0.05.

## Results

### Ex vivo mouse IgG ELISA, quantification of respiratory burst, CFU and biofilm biomass

Systemic IgG levels as measured by an IgG ELISA assay were significantly higher in the blood of CIA mice ($3.1 \times 10^2$ +/- $2.6 \times 10^1$ ng/mL) as compared to unimmunized, control mice ($1.9 \times 10^2$ +/- 8.6 ng/mL; p = 0.003) (Fig 1).

Respiratory burst, as measured by mean fluorescent intensity from a DHR 123 assay, was not significantly different between CIA mice ($4.9 \times 10^4$ +/- $2.6 \times 10^3$) and control mice ($4.9 \times 10^4$ +/- $2.4 \times 10^3$; p = 0.89) (Fig 2).

The mean CFU/mL of *S. aureus Xen36* after interaction with whole blood in the CIA group ($2.6 \times 10^3$ +/- $1.7 \times 10^2$ CFU/mL) was not significantly different than control mice ($2.6 \times 10^3$ +/- $2.4 \times 10^2$ CFU/ml; p = 0.91) (Fig 3).

Residual *S. aureus* biofilm after mixture of bacteria and blood, as measured by absorbance units, was not significantly different between the CIA group (1.3 +/- 0.3) and the control group (1.2 +/- 0.3; p = 0.96) (Fig 4).

### In vivo quantification of bacterial burden using longitudinal bioluminescent imaging

There was no significant difference in bioluminescent signal between the CIA group and the infected control group at all postoperative time points. Immediately following surgery (POD 0), the CIA group ($2.0 \times 10^4$ +/- $1.4 \times 10^3$ photons/s/cm$^2$/sr) showed similar bioluminescence as the infected control group ($1.9 \times 10^4$ +/- $2.0 \times 10^3$ photons/s/cm$^2$/sr; p = 0.92). Peak bioluminescent signal occurred at POD 3 in both the CIA group ($4.0 \times 10^5$ +/- $3.5 \times 10^4$ photons/s/cm$^2$/sr) and the infected control group ($3.9 \times 10^5$ +/- $5.6 \times 10^4$ photons/s/cm$^2$/sr; p = 0.93). Immediately prior to sacrifice on POD 28, the signal in the CIA group ($7.9 \times 10^4$ +/- $1.4 \times 10^4$ photons/s/cm$^2$/sr) was similar to the infected control group ($8.0 \times 10^4$ +/- $2.2 \times 10^4$; p = 0.92) (Fig 5A and 5B).

### Confirmation of bacterial burden using implant and surrounding tissue CFU quantification

Viable *S. aureus* Xen36 CFUs were identified in 0% (0 of 2) of implants from the sterile group, 62.5% (5 of 8) of implants in the infected control group and 58.3% (7 of 12) of implants in the CIA group. No significant difference existed between the mean CFUs/mL adherent to the harvested implants in the CIA group ($8.1 \times 10^2$ +/- $3.4 \times 10^2$) and the infected control group ($1.0 \times 10^3$ +/- $6.0 \times 10^2$; p = 0.82) (Fig 6A). Similarly, no significant difference existed between the mean CFUs/mL from the harvested tissue in the CIA group ($1.1 \times 10^4$ +/- $9.1 \times 10^3$) and the infected control group ($1.4 \times 10^4$ +/- $9.6 \times 10^3$; p = 0.80) (Fig 6B).

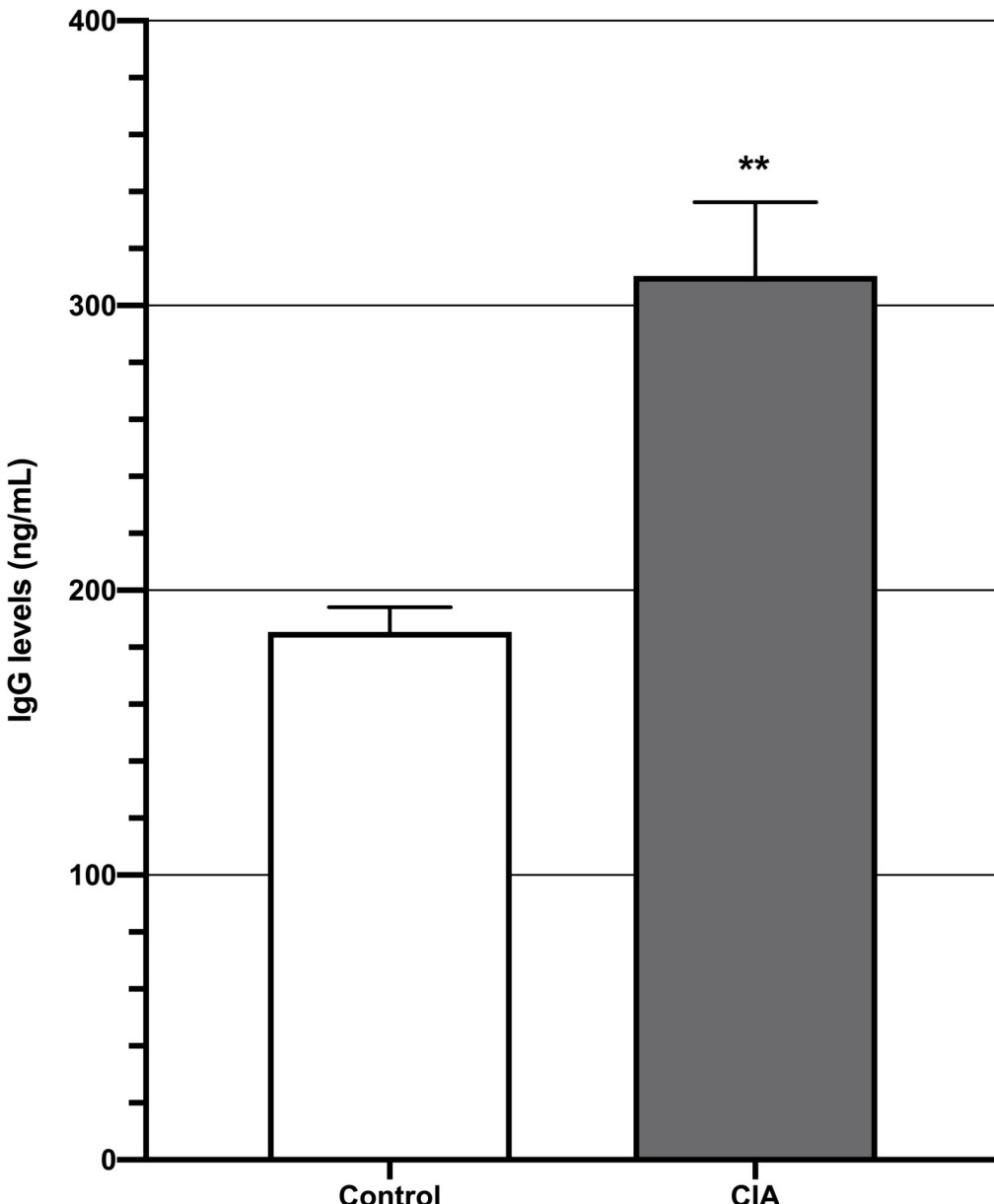

**Fig 1. ELISA demonstrating that mice with CIA had significantly higher systemic IgG levels than control mice 6 weeks after induction of CIA.** ** denotes p<0.01.

## CIA severity score

The mean clinical severity scores of CIA mice at each postoperative time point ranged from 5.5 to 6.25 with a range of 5 to 8 for each individual mouse (Fig 7A–7C).

## Discussion

Due to articular chondrolysis, RA often results in debilitating joint pain, thus driving the need for arthroplasty in this patient population [18, 23, 24]. PJI is a devastating complication of total

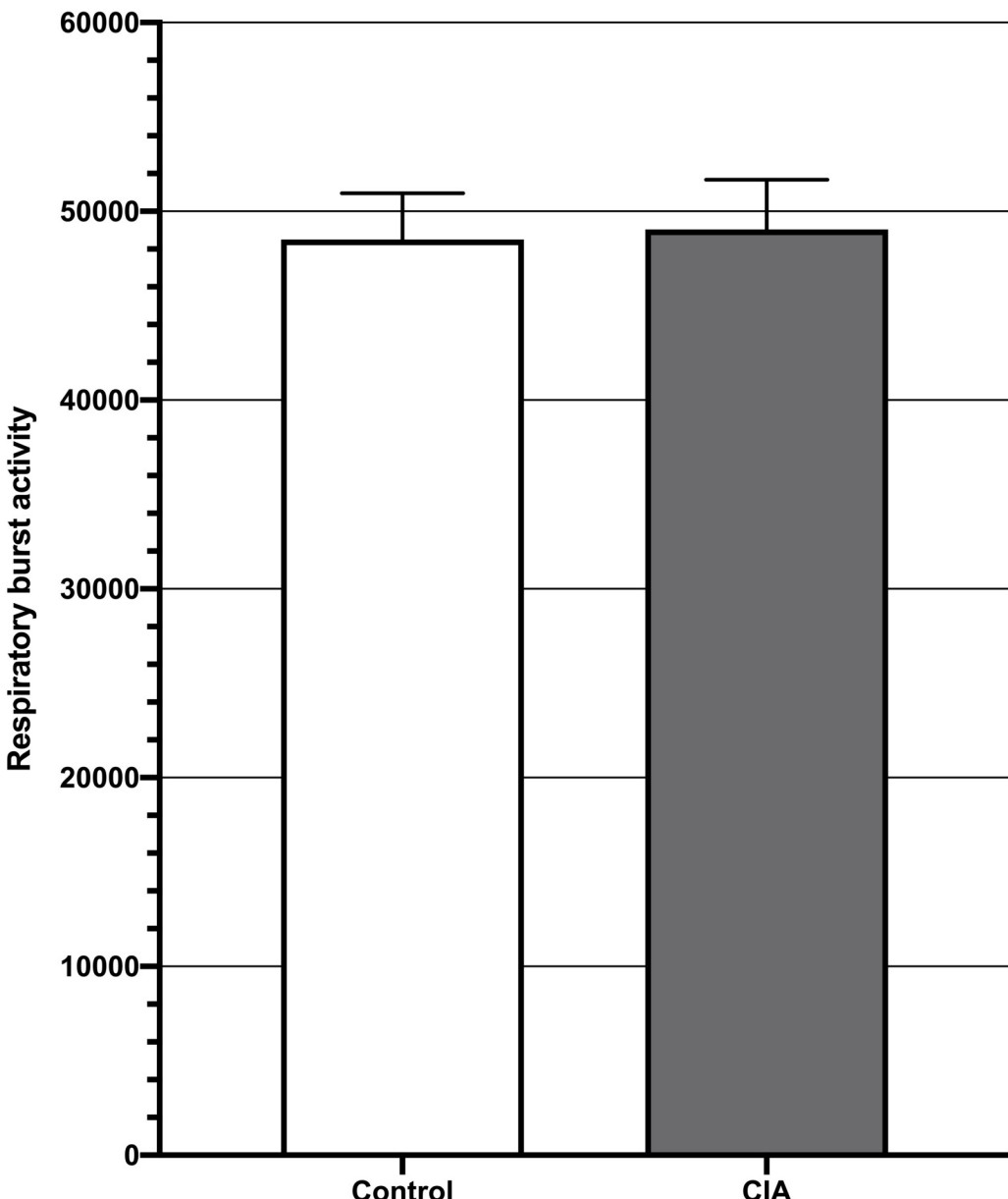

**Fig 2. Measurement of respiratory burst *ex vivo* showing no significant difference in respiratory burst between the blood of mice with CIA when compared to control mice (p = 0.89).**

joint arthroplasty with poor outcomes, high patient morbidity and mortality as well as limited treatment options [52–54]. Current literature to date is inconsistent with regard to whether RA is an inherent risk factor for PJI. Given the health and economic burden of PJI to the United States healthcare system [4–6, 9], it is of the utmost importance to thoroughly study factors that affect infectious risk. There is currently an unmet need to thoroughly investigate the impact that RA alone, without therapeutics, has on the risk of PJI.

## *Ex vivo* CFU Quantification

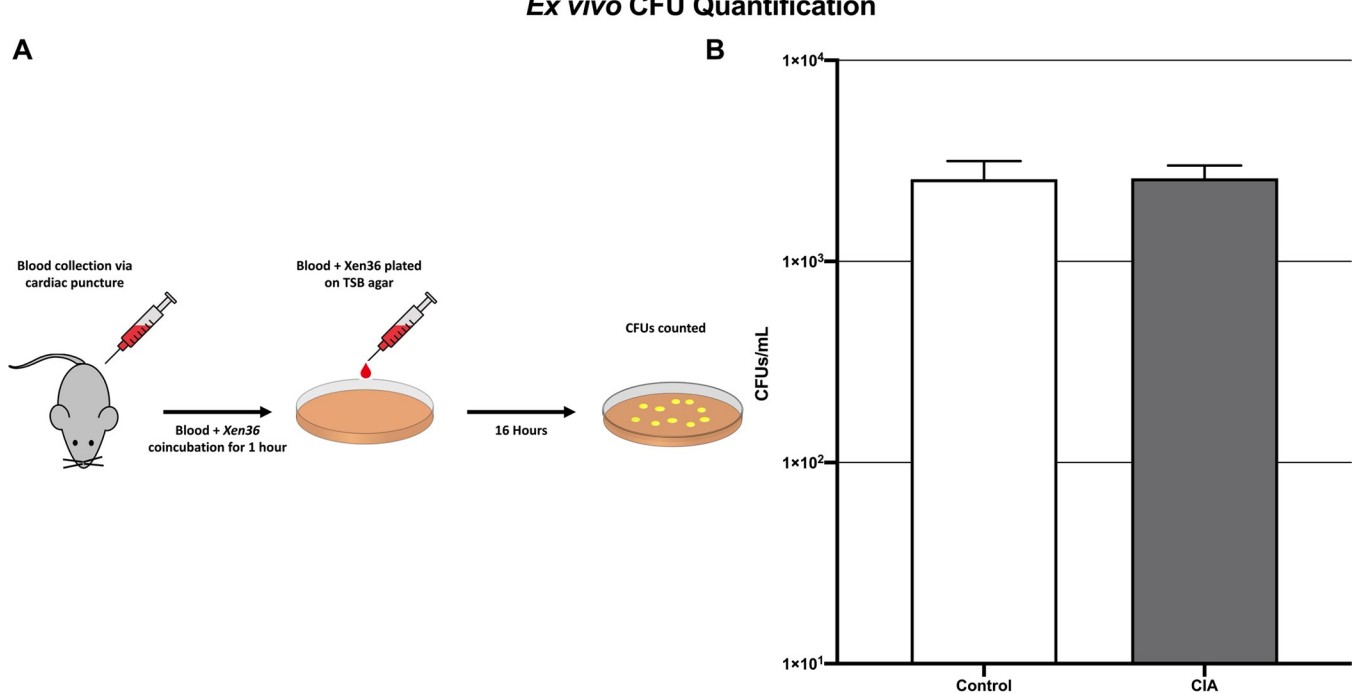

**Fig 3.** Blood collection and plating procedure (A) and subsequent CFU counts demonstrating no significant difference in *S. aureus* burden *ex vivo* in the blood of mice with CIA as compared to control mice (B) ($p = 0.91$).

## *Ex vivo* Measurement of Biofilm Formation

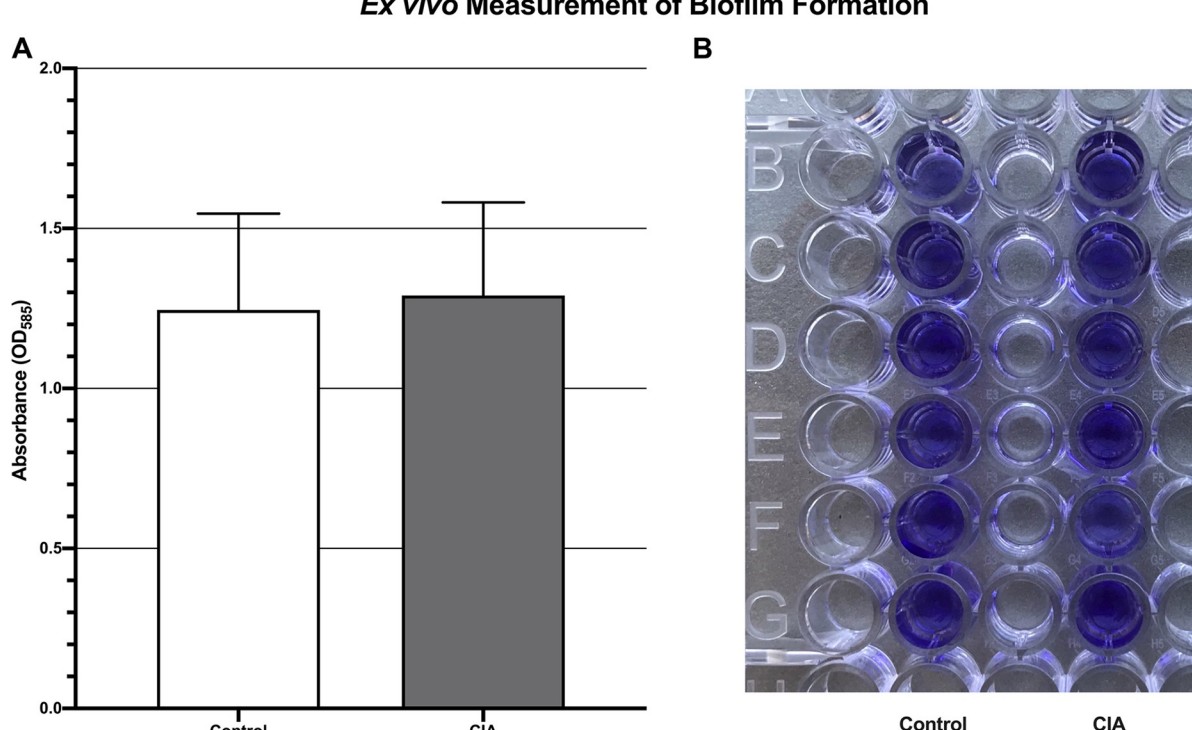

**Fig 4.** *Ex vivo* quantification of biofilm formation demonstrating no significant difference in *S. aureus* Xen36 burden between the blood of mice with CIA and control mice (A). This was measured by absorbance after staining of residual biofilm with crystal violet (B) ($p = 0.96$).

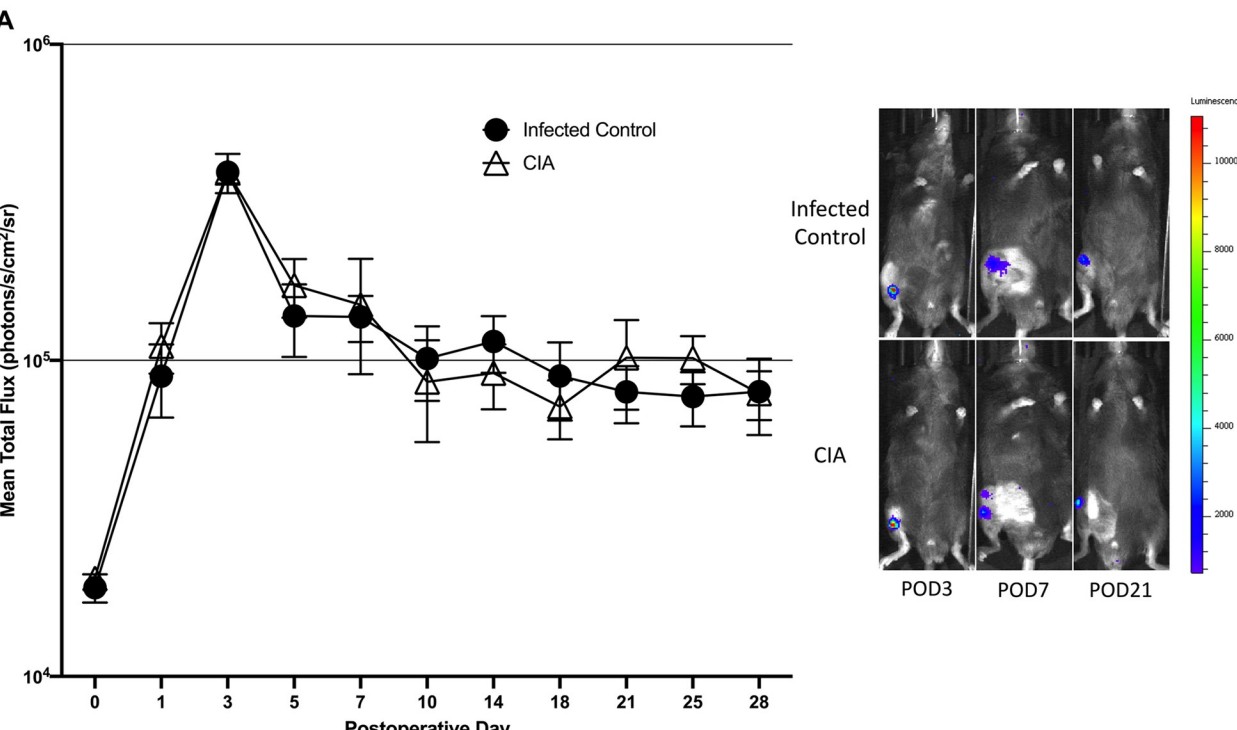

**Fig 5.** *S. aureus* burden *in vivo* demonstrating no significant difference between mice with CIA and infected control mice at all postoperative time points (A). Representative images depicting *in vivo S aureus* Xen36 bioluminescence at three postoperative time points (B).

In this study, a well-validated animal model of human RA, CIA, along with a preclinical model of PJI was combined in a novel fashion to investigate if RA correlates with infectious risk after arthroplasty. CIA mice utilized in this study demonstrated paw edema scores between 5 to 8, indicating that they were clinically representative of moderately severe RA. Systemic alteration of the immunologic profile of CIA mice was confirmed as CIA mice demonstrated significantly elevated levels of systemic IgG as compared to control mice. The blood of CIA mice demonstrated similar respiratory burst capacity to the blood of control mice. Furthermore, *ex vivo S. aureus* growth and biofilm inhibition did not significantly differ in the CIA mice as compared to infected control mice. CIA mice also demonstrated no significantly different *S. aureus* burden *in vivo* compared to infected control mice as bioluminescent signals started, peaked and ended at remarkably similar levels at POD 0, 3 and 28, respectively. Additionally, 58.3% of implants in CIA mice showed viable, adherent *S. aureus* as compared to 62.5% of implants in the infected control group. Concurrent with these findings, implant and surrounding tissue *S. aureus* CFU counts 28 days after surgery and inoculation did not significantly differ between the groups. Thus, although CIA mice had an altered immune profile, no difference in *S. aureus* burden was appreciated both *ex-vivo* and *in-vivo* when compared to control mice.

Taken together, the results of this study suggest the possibility that moderately severe RA alone may not represent an inherently significant infectious risk factor. As such, the perioperative management of patients with moderately severe RA may not have to be based on concern for infectious risk. When considering the perioperative infectious burden of patients with RA,

## CFUs Harvested From Implant and Tissue *in vivo*

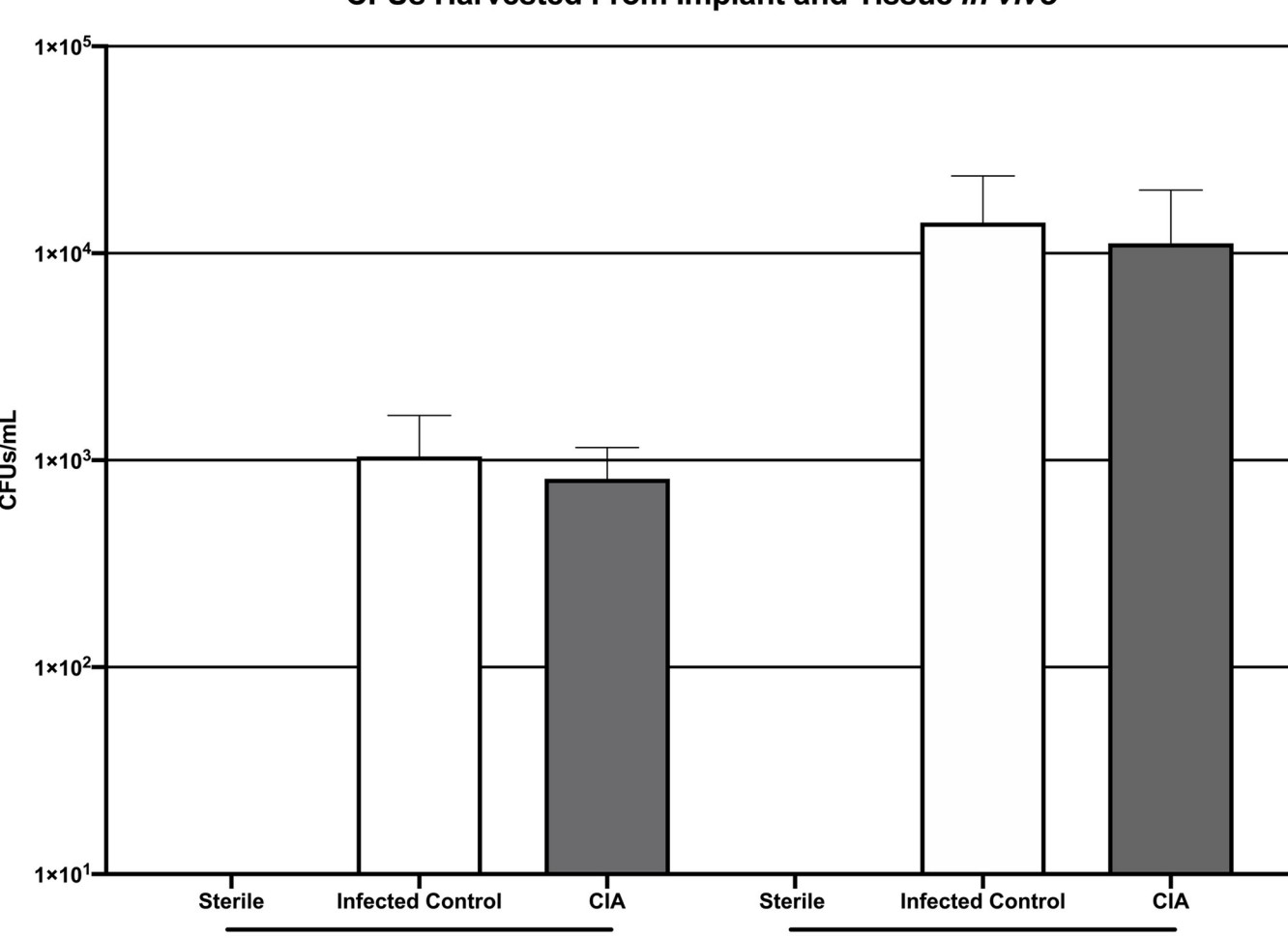

**Fig 6.** CFU quantification of *S. aureus* harvested from implants (A) and surrounding tissue (B) demonstrating no significant difference in *S. aureus* burden between mice with CIA and infected control mice (p = 0.82 and p = 0.80 for implant and tissue, respectively).

it is important to consider how the upregulation of the immune response of RA weighs against its general dysregulation. If a clinical association between RA and PJI in fact exists, as has been suggested in the literature [24, 28–33], it is possible that the cause of this association may be secondary to more modifiable factors, such as the perioperative use of immunosuppressive medications and secondary wound complications, rather than the disease process itself. Isolating the infectious burden of RA, alone, is the first step in developing strategies and guidelines to mitigate the risk of PJI in this population. Once this infectious risk is understood, factors such as perioperative immunomodulatory medication management as well as wound healing complications can be thoroughly investigated.

There are several limitations to this study and further translational and clinical investigation is certainly warranted. Although CIA is the gold standard animal model for studying RA, as CIA mice generate autoantibodies to collagen in a similar fashion to humans with RA, the exact immune profile of human RA remains difficult to perfectly duplicate. As such, the immune response in CIA mice may not perfectly replicate the immune response in human RA. Furthermore, the current study is not suited to investigate the mechanistic molecular

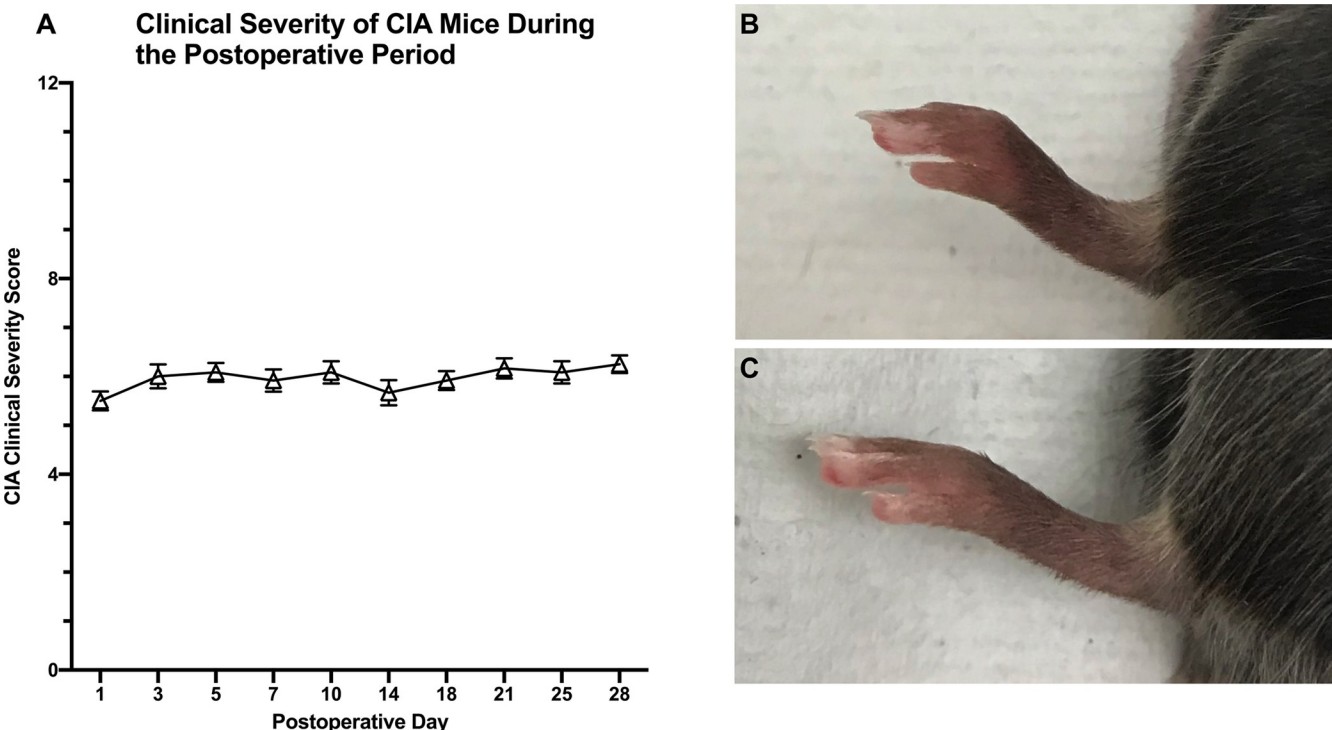

**Fig 7.** Mean clinical severity score of mice induced with CIA demonstrating a consistent level of moderate arthritis throughout the study period (A). Scores for each paw were given on a scale of 0–3 based on the amount of dorsal edema and added together for a total maximum score of 12. Representative images of paws of mice with CIA (B) and normal mice (C) are shown.

rationale for the findings that no increased infectious burden was seen in CIA mice. As such, this study cannot comment on how the dysregulated immune system of CIA responded to inoculation nor how it affected biofilm formation. This study also only investigated the immune response of mice that had moderately severe CIA. Although this level of severity was desired as it is moderately severe RA that causes the most uncertainty regarding optimal peri-operative management, these results cannot be extrapolated to mild or severe RA. In this study, no subjective difference in wound healing was noted between experimental groups. However, wound healing in this mouse model of PJI is an imperfect representation of that which occurs following arthroplasty in humans. This is an important limitation, as patients with RA are generally reported to have higher rates of wound complications, increasing their susceptibility to PJI [37, 55, 56]. Another limitation to this study is that systemic IgG levels were measured prior to surgery to confirm a molecular change in CIA mice. IgG levels were not measured after surgery and inoculation; however the aim of this study was not to analyze how these IgG levels would be affected with the introduction of bacteria. Lastly, this implant infection model is a simplification of the steps involved in joint arthroplasty, so the transla-tional limitations of this model should be considered. The advantages of this model are widely viewed to outweigh the accepted limitations as this model allows for a safe, reproducible and well-powered manner to longitudinally quantify infectious burden *in vivo*.

Despite these limitations, this study represents the first of its kind to examine the effect of RA on infectious burden in PJI using a novel combination of two well-validated animal mod-els. The current findings provide *ex vivo* and *in vivo* evidence that mice with CIA do not

demonstrate any significant difference in susceptibility to *S. aureus* infection as compared to control mice. These results suggest that moderately severe RA, alone, may not pose significant postoperative infectious risk. Given the translational nature of this mouse model, this study addresses an unmet need by adding to the current understanding of how RA affects PJI risk in a novel fashion. This study can lead to further clinical studies with the ultimate goal of developing strong strategies and guidelines to mitigate postoperative infections in patients with RA undergoing arthroplasty.

## Supporting information

**S1 Checklist. ARRIVE (Animal Research: Reporting of In Vivo Experiments) essential 10 items to assess the reliability of findings for manuscripts that included in vivo experiments.**
(PDF)

## Author Contributions

**Conceptualization:** Rishi Trikha, Danielle Greig, Troy Sekimura, Nicolas Cevallos, Benjamin Kelley, Zeinab Mamouei, Christopher Hart, Amr Turkmani, Adam Sassoon, Alexandra Stavrakis, Nicholas M. Bernthal.

**Data curation:** Rishi Trikha, Danielle Greig, Troy Sekimura, Nicolas Cevallos, Christopher Hart, Micah Ralston, Amr Turkmani, Adam Sassoon, Alexandra Stavrakis, Nicholas M. Bernthal.

**Formal analysis:** Rishi Trikha, Benjamin Kelley, Adam Sassoon, Nicholas M. Bernthal.

**Funding acquisition:** Nicholas M. Bernthal.

**Investigation:** Rishi Trikha, Danielle Greig, Troy Sekimura, Nicolas Cevallos, Zeinab Mamouei, Micah Ralston, Amr Turkmani, Alexandra Stavrakis, Nicholas M. Bernthal.

**Methodology:** Rishi Trikha, Troy Sekimura, Micah Ralston, Nicholas M. Bernthal.

**Project administration:** Rishi Trikha, Troy Sekimura, Benjamin Kelley, Zeinab Mamouei, Micah Ralston, Amr Turkmani.

**Resources:** Nicholas M. Bernthal.

**Software:** Rishi Trikha.

**Supervision:** Rishi Trikha, Adam Sassoon, Alexandra Stavrakis, Nicholas M. Bernthal.

**Validation:** Rishi Trikha, Adam Sassoon, Alexandra Stavrakis, Nicholas M. Bernthal.

**Visualization:** Rishi Trikha, Micah Ralston, Adam Sassoon, Alexandra Stavrakis, Nicholas M. Bernthal.

**Writing – original draft:** Rishi Trikha, Danielle Greig, Troy Sekimura, Nicolas Cevallos, Benjamin Kelley, Zeinab Mamouei, Christopher Hart, Micah Ralston, Amr Turkmani, Adam Sassoon, Alexandra Stavrakis, Nicholas M. Bernthal.

**Writing – review & editing:** Rishi Trikha, Danielle Greig, Troy Sekimura, Nicolas Cevallos, Benjamin Kelley, Zeinab Mamouei, Christopher Hart, Alexandra Stavrakis, Nicholas M. Bernthal.

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
