## [Decision Letter · Decision Letter 0]

10 Jun 2021

PONE-D-21-09243

Active Rheumatoid Arthritis in a Mouse Model is not an Independent Risk Factor for Periprosthetic Joint Infection

PLOS ONE

Dear Dr. 

Thank you for submitting your manuscript to PLOS ONE. After careful consideration, we feel that it has merit but does not fully meet PLOS ONE’s publication criteria as it currently stands. Therefore, we invite you to submit a revised version of the manuscript that addresses the points raised during the review process.

We look forward to receiving your revised manuscript.

Kind regards,

Rosanna Di Paola, MD

Academic Editor

PLOS ONE

Journal Requirements:

2. As part of PLOS ONE's publication criteria, the journal requires that in each submission, experiments, statistics, and other analyses are performed to a high technical standard and are described in sufficient detail (https://journals.plos.org/plosone/s/criteria-for-publication). In the case of your paper, please revise your Methods section to address the following: (1) the number of animals in each group and how you determined the sample size; (2) the sex and strain of the mice; (3) if applicable: all anesthetics and analgesics administered to animals during your study (name of drug, dosage, frequency and route of administration); (4) details about humane endpoints for any animals who became severely ill during the study; (5) the rate of mortality during the study and the cause of death (if applicable); (6) the criteria used to determine when to euthanize animals (for animals who became ill prior to the experimental endpoints); (7) the method of euthanasia.

Additional Editor Comments:

Reviewers' comments:

Reviewer's Responses to Questions

**Comments to the Author**

1. Is the manuscript technically sound, and do the data support the conclusions?

Reviewer #1: Yes

Reviewer #2: Yes

2. Has the statistical analysis been performed appropriately and rigorously? 

Reviewer #1: Yes

Reviewer #2: Yes

3. Have the authors made all data underlying the findings in their manuscript fully available?

Reviewer #1: Yes

Reviewer #2: Yes

4. Is the manuscript presented in an intelligible fashion and written in standard English?

Reviewer #1: Yes

Reviewer #2: Yes

5. Review Comments to the Author

Reviewer #1: This study investigated the mechanism underlining periprosthetic joint infection in active rheumatoid arthritis. The rational behind the experiment was clear and straight forward. The manuscript is almost well written

While many different sources are used to set up the study in the introduction, little previous evidence is

stated. The introduction is thus short and poorly sets up the rationale for the study. More attention to how

this study fits into previous work in rheumatoid arthritis and inflammation should be added to improve this section. Please refer to doi: 10.3390/antiox9060511, 10.1186/s13075-019-2048-y.

There are some minor grammar issues that should be fixed in order to aid the accessibility of the results to

the reader.

Reviewer #2: The submission from Rishi Trikha et al. reports that active RA in a mouse model is not an independent Risk Factor for Periprosthetic Joint Infection. The study is interesting and well structured. However, there are some corrections.

Minor comments:

1. Please refer to doi: 10.1016/j.biopha.2020.110023; 10.3390/app10041324

2. The authors should include in the materials and methods section whether the mice used were male or female.

3. The authors should better highlight the purpose of the study and the novelty

4. The authors should better check the manuscript for any typographical errors.

6. PLOS authors have the option to publish the peer review history of their article (what does this mean?). If published, this will include your full peer review and any attached files.

Reviewer #1: No

Reviewer #2: No

---

## [Author Response · Author response to Decision Letter 0]

22 Jul 2021

The response letter has been uploaded, however please also see below

Dear Editorial Team at the Public Library of Science,

Thank you for taking the time to offer your suggestions. We truly appreciate your input and will address any/all concerns. Please see our responses to each reviewer below. All changes are line referenced.

Reviewer #1: 

This study investigated the mechanism underlining periprosthetic joint infection in active rheumatoid arthritis. The rationale behind the experiment was clear and straight forward. The manuscript is almost well written

While many different sources are used to set up the study in the introduction, little previous evidence is stated. The introduction is thus short and poorly sets up the rationale for the study. More attention to how this study fits into previous work in rheumatoid arthritis and inflammation should be added to improve this section. Please refer to doi: 10.3390/antiox9060511, 10.1186/s13075-019-2048-y. There are some minor grammar issues that should be fixed in order to aid the accessibility of the results to

the reader.

Thank you for these astute comments. In an effort to better elucidate the rationale as well as the novelty of our study we have expanded the introduction and added to the discussion. 

Lines 74-79, 115-117, 122-23 have been added to expand upon the importance of this study as well as the prevention of PJI in general.

Lines 81-84 have been reorganized and replaced at the beginning of the second paragraph of the introduction to ensure that the rationale of our study is better understood. 

The authors have added text regarding the molecular basis of rheumatoid arthritis as well as novel therapeutics countering synovial inflammation referenced from both of the intriguing papers suggested (Lines 87-109).

The authors have further expanded on the molecular basis of collagen-induced arthritis (Lines 126-128).

The authors believe that the concluding sentence of the introduction now better highlights the rationale of our study (Lines 160-162).

The authors have added to and restructured the end of the discussion to ensure that the novelty, significance and rationale of the study are clearer (Lines 437-443).

The authors have made grammatical corrections to ensure accessibility.

Reviewer #2: The submission from Rishi Trikha et al. reports that active RA in a mouse model is not an independent Risk Factor for Periprosthetic Joint Infection. The study is interesting and well structured. However, there are some corrections.

Minor comments:

1. Please refer to doi: 10.1016/j.biopha.2020.110023; 10.3390/app10041324

2. The authors should include in the materials and methods section whether the mice used were male or female.

3. The authors should better highlight the purpose of the study and the novelty

4. The authors should better check the manuscript for any typographical errors.

1/3. Thank you for these insightful comments, we have referenced the article by Impellizzeri et al. to further explain the molecular basis of rheumatoid arthritis (Lines 87-88) as well as expand upon the molecular basis of collagen-induced arthritis (Lines 125-128). Furthermore, we have explained that while novel therapeutics are being studies (as referenced by Sunzini et al. along with two other studies), little evidence regarding the infectious burden of rheumatoid arthritis exists (Lines 107-108). 

The authors believe that the purpose of this study has been set-up in a more coherent fashion by inserting/replacing the first sentence of the second paragraph of the introduction (Lines 81-84). The authors also believe that the novelty and significance of this study has been made clearer by incorporating the last sentence of the introduction (Lines 160-162) as well as the last three sentences of the discussion (Lines 437-443).

2. The authors have clarified that the mice used were male (Line 210); thank you for pointing this out.

4. The authors have made grammatical corrections to ensure accessibility.

---

## [Editor Report · Decision Letter 1]

30 Jul 2021

Active Rheumatoid Arthritis in a Mouse Model is not an Independent Risk Factor for Periprosthetic Joint Infection

PONE-D-21-09243R1

Dear Dr. Trikha,

We’re pleased to inform you that your manuscript has been judged scientifically suitable for publication and will be formally accepted for publication once it meets all outstanding technical requirements.

Kind regards,

Rosanna Di Paola, MD

Academic Editor

PLOS ONE
---

## [Editor Report · Acceptance letter]

6 Aug 2021

PONE-D-21-09243R1 

Active Rheumatoid Arthritis in a Mouse Model is not an Independent Risk Factor for Periprosthetic Joint Infection 

Dear Dr. Trikha:

I'm pleased to inform you that your manuscript has been deemed suitable for publication in PLOS ONE. Congratulations! Your manuscript is now with our production department. 

Kind regards, 

on behalf of

Dr. Rosanna Di Paola 

Academic Editor

PLOS ONE